Immune responsiveness to phytohemagglutinin displays species but not sex differences in three anuran species

Zhang Zhiqiang zzq-003@163.com 1
Jin Chenchen 2
Qu Kangshan 1
Caviedes-Vidal Enrique 3 4
1 College of Animal Science and Technology, Anhui Agricultural University , Hefei , People’s Republic of China
2 College of Life Science, Anhui Agricultural University , Hefei , People’s Republic of China
3 Instituto Multidisciplinario de Investigaciones Biológicas de San Luis, Consejo Nacional de Investigaciones Científicas y Técnicas, and Departamento de Bioquímica y Ciencias Biológicas, Universidad Nacional de San Luis , San Luis , Argentina
4 Department of Forest and Wildlife Ecology, University of Wisconsin-Madison , Madison , United States
Pie Marcio
Electronic publication date: 2017 May 3
Publication date: 2017
Volume: 5
Electronic Location ID: e3181
Received 2016 Oct 18; Accepted 2017 Mar 14
Copyright: ©2017 Zhang et al.
Copyright year: 2017
Copyright holder: Zhang et al.
License: This is an open access article distributed under the terms of the Creative Commons Attribution License, which permits unrestricted use, distribution, reproduction and adaptation in any medium and for any purpose provided that it is properly attributed. For attribution, the original author(s), title, publication source (PeerJ) and either DOI or URL of the article must be cited.
License URL: https://creativecommons.org/licenses/by/4.0/

Keywords: Immunity, PHA assay, Anurans, Species, Sex, Pseudepidalea raddei, Pelophylax nigromaculatus, Bufo gargarizans

Funding: National Natural Science Foundation of China 31170379 Stable and Excellent Talent Plan of Anhui Agricultural University 2010-WD-02 This study was supported by the National Natural Science Foundation of China (No. 31170379) and the Stable and Excellent Talent Plan of Anhui Agricultural University (No. 2010-WD-02). The funders had no role in study design, data collection and analysis, decision to publish, or preparation of the manuscript.

==============================
Phytohemagglutinin (PHA)-induced skin swelling response is widely used as a rough surrogate of integrative cell-mediated and innate immunity across multiple vertebrate taxa due to its simplification and feasibility. However, little is known whether there are sex and interspecific differences of immune responsiveness to PHA in ectotherms, especially for anurans. Therefore, we studied sex and species differences of PHA response in three anurans, Asiatic toads (Bufo gargarizans), Dark-spotted frogs (Pelophylax nigromaculatus) and Mongolian toads (Pseudepidalea raddei), captured in northern regions of Anhui Province (China). Footpad thickness was measured prior to (0 h) and after (6, 12, 24, 48 and 72 h) a PHA injection and normalized against saline injection in the opposite footpad. Body mass was recorded at the beginning (0 h) and end of each assay (72 h). Results showed effects of PHA assay, sex and taxa on body mass. Relative maximum swelling response (PHAmax) ranged from 18.58–29.75%, 9.77 to 20.56% and 21.97 to 31.78% and its occurrence over time was apparent 10.6–19.72 h , 7.74–14.01 h and 17.39–23.94 h postinjection for Asiatic toads, Dark-spotted frogs and Mongolian toads, respectively. Finally, the magnitude or timing of PHAmax in Dark-spotted frogs was significantly thinner and faster than in Mongolian toads, and Asiatic toads had an in-between value, not different from the other two species. The magnitude of PHAmax was significantly positively correlated with the timing of PHAmax considering individuals altogether, but not when analyzed within species. Our results indicate that male and female anuran species respond similarly to PHA antigen stimulation, but the magnitude and timing of PHAmax is species-specific. Briefly, we provide new evidence for the suitability of PHA assay in non-model anuran species with different body sizes, and exhort the need to further investigate the nature of PHA assay at the hematological and histological levels in order to extend its application in ecoimmunological studies of amphibians.

Introduction

Amphibians are experiencing a global biodiversity crisis derived from diseases, habitat destruction and altered ecosystems (Clulow, Harris & Mahony, 2015). A wide range of physiological traits including innate and adaptive immunity are important to cope with the increasing number of stressors and pathogen threats faced in their lifetimes (Carey, Cohen & Rollins-Smith, 1999). As other physiological traits, immune defense is also energetically costly to mount, and a trade-off relationship may occur within the immune system or between immunological defenses and other nutrient-demanding processes such as growth, reproduction and thermoregulation based on limited total energy resources (Lochmiller & Deerenberg, 2000). Nowadays, it is still an open question what immune parameter measures are appropriate to reflect an organism’s defense capacity against pathogens (Downs, Adelman & Demes, 2014). Among several immunological assays, the phytohemagglutinin (PHA) skin-swelling test is one of the most widely used techniques to evaluate the magnitude of vertebrate immune response (Sheldon & Verhulst, 1996; Boughton, Joop & Armitage, 2011; Vinkler & Albrecht, 2011; Brock, Murdock & Martin, 2014). The assay consists of a subcutaneous injection of PHA that induces infiltration of leukocytes and where swelling is interpreted as an integrative immune response (Christe, Arlettaz & Vogel, 2000; Kennedy & Nager, 2006; Martin et al., 2006; Demas et al., 2011; Brock, Murdock & Martin, 2014) or as an inflammation in endotherms (Vinkler et al., 2012; Bílková, Vinklerová & Vinkler, 2015; Bílková et al., 2016). However, inferences using this assay should be carefully evaluated because not always a greater immune response or inflammation means better defenses against infections or higher survival probabilities (Graham, Allen & Read, 2005; Vinkler et al., 2012). To date, the histological base of this response has been demonstrated across different vertebrate taxa (e.g., birds (Martin et al., 2006), bats (Turmelle et al., 2010), amphibians (Brown, Shilton & Shine, 2011) and crocodiles (Finger et al., 2013)), although no apparent correlation between swelling response and leukocyte profiles was reported in the fossorial mammal Ctenomys talarum Merlo, Cutrera & Zenuto, 2014). However, more recently PHA swelling has been associated to leukocyte profiles and serum bactericidal activity in endotherms (Bílková, Vinklerová & Vinkler, 2015; Zhang & Zhao, 2015), or parasite infection in rodents (Ctenomys talarum, Merlo, Cutrera & Zenuto, 2015). Despite all the merits of this assay, a caveat must be warned because there is still some debate about its meaning. An objection is that the response to PHA of an organism might not be representative of the actual response of the same host to pathogens (reviewed by Graham, Allen & Read, 2005; Owen & Clayton, 2007; Brock, Murdock & Martin, 2014) or that energetic costs of mounting a response to PHA may vary in small mammals (Zhang, Qiu & Wang, 2011; Merlo et al., 2014) and birds (Ots et al., 2001; Martin, Scheuerlein & Wikelski, 2003; Lee, Martin & Wikelski, 2005; Nilsson, Granbom & Raberg, 2007).

PHA assay was first used in ecoimmunological studies of amphibians in the northern leopard frogs, Rana pipiens, showing noticeable alterations of the immune response after an exposure to different pesticides (Gilbertson et al., 2003). The sites chosen to inject PHA are diverse for post-metamorphic amphibians and none is applicable to all species, for example, toe webbing (Brown, Shilton & Shine, 2011), footpad (Fites et al., 2014), thigh (Gervasi & Foufopoulos, 2008) and lower leg below the knee (Clulow, Harris & Mahony, 2015). The histological mechanism of the inflammatory response induced by PHA antigen was examined in the toe webbing of cane toads, Rhinella marinus (Brown, Shilton & Shine, 2011), which displayed a rapid (<12  h) infiltration of neutrophils, eosinophils and macrophages at the injection site, followed by an influx of lymphocytes by 24 h postinjection. In addition, marked interspecific variations of leukocytes were observed in the lower leg of green and golden bell frogs, Litoria aurea and striped marsh frogs, Limnodynastes Peronii (Clulow, Harris & Mahony, 2015). PHA assay has been also used to assess the immune responsiveness under different ecologically relevant conditions, such as when amphibians are forced to accelerate the metamorphosis by exposition to environment desiccation (Gervasi & Foufopoulos, 2008), treated with protein restricted diets (Venesky et al., 2012), and exposed to density stress (Clulow, Harris & Mahony, 2015). In different vertebrate species, PHA response peaks in a wide time window, from 3 to 48 h (De Bellocq et al., 2006; Martin et al., 2006; Turmelle et al., 2010; Xu & Wang, 2010; Brown, Shilton & Shine, 2011; Finger et al., 2013; Josserand et al., 2015; Zhang, Huang & Zhao, 2015; Bílková et al., 2016), with no apparent sex differences in some small mammals (Zhang, Qiu & Wang, 2011; Merlo, Cutrera & Zenuto, 2014) and amphibians (Brown et al., 2015; Clulow, Harris & Mahony, 2015). Most up-to-date information available on the time course and temporal dynamics of the PHA responses within and among populations or species are for birds (Navarro et al., 2003) and small mammals (De Bellocq et al., 2006; Zhang, Huang & Zhao, 2015). Conversely, data on amphibians are still scarce, especially for those anurans living in different environments (Clulow, Harris & Mahony, 2015). In a scenario of increasing disease and toxic threats, it is mandatory to provide comparable information of individual health status induced by PHA antigens on key amphibian species (Carey, Cohen & Rollins-Smith, 1999; Fites et al., 2014; Clulow, Harris & Mahony, 2015).

In line with this situation, our aim was to test PHA assay response across different species with different body masses to extend its use to other non-model anuran species with different morphological and physiological adaptations to their habitats. Accordingly, we provide an evaluation of the PHA-induced skin-swelling assay among three sympatric anuran species with different habitat use in a range of body mass (30–90 g) of Central China (Northern Anhui province), Dark-spotted frogs (Pelophylax nigromaculatus), Mongolian toads (Pseudepidalea raddei) and Asiatic toads (Bufo gargarizans). In the three anuran species, both Asiatic and Mongolian toads are terrestrial with slow movements, whereas Dark-spotted frogs dwell in lentic ponds or rice fields and move fast in their habitats (Fei et al., 2009a; Fei et al., 2009b). In addition, Asiatic toads start their reproductive activities during hibernation (Ji et al., 1995; Zou et al., 1996) and are easily infected by multiple parasites (Zhao et al., 2014). On the other side, while Mongolian toads (Zhou, 1996) and Dark-spotted frogs initiate their reproduction after hibernation (Wu, 1965; Cao et al., 2000; Jin, Qu & Zhang, 2014; Jin & Zhang, 2014), the former is not infected by multiple digenetic trematodes (Li & Gu, 1978), but the latter, displays a 90.38% helminth and parasite infection rate (Zhu et al., 2014). We predicted that: (i) Within species, both sexes will exhibit similar swelling responses. (ii) The temporal dynamics of the response to PHA will be species-specific, as well as the magnitude of the maximum swelling at the injection point in the footpad area. In addition, we tested the effect of the PHA assay on the body mass of the individuals and its influence on the swelling response.

Materials and Methods

Animal husbandry

All animal procedures were carried out under the guidelines of the Animal Care and Use Committee of Anhui Agricultural University (Permit number: 20120410). During April and May, 2012, we captured nine and eight male and nine and ten female Asiatic and Mongolian toads, respectively, in Mengcheng county (116.55°E 33.25°N Anhui province, China), and eight male and eight female Dark-spotted frogs in Feixi county (117.17°E 31.70°N Anhui province, China). Immediately after capture, animals were transported to an animal indoor facility of the Anhui Agricultural University in Hefei, and housed in aquarium tanks (40 × 50 × 90 cm). They were kept under natural ambient temperature (17−21°C) and photoperiod (around 12L:12D). Fresh dechlorinated tap water was provided ad libitum and mealworms were provided every 2–3 days during trials.

PHA-induced skin-swelling test

In all cases PHA-induced skin-swelling responses were always assayed two or three days after capture. Animals were subcutaneously injected with a dose of 50 µl of the PHA solution (PHA-P, Sigma L-8754, Sigma-Aldrich, St Louis, MO, USA, 1 mg of PHA-P dissolved in 0.3 mL of saline solution) in the footpad area pointing to the junction of the second and third digits of the right hind foot, and the opposite left footpad as an intragroup control was injected with an equal volume of sterilized saline by using a 100 µl micro injector (Shanghai Anxiang micro injector factory, China). Footpad thickness was measured with a micrometer (Tesa Shopcal, Renens, Switzerland) to ±0.01 mm, prior to injection (time = 0 h ), and then at 6, 12, 24, 48 and 72 h postinjection. Footpad thickness was measured six times at each time point by the same operator at the point of injection and the average value was used for analysis (Smits, Bortolotti & Tella, 1999; De Bellocq et al., 2006). Body mass (±0.001 g) was recorded at time 0 h and then, 72 h after injection in each individual of each species. In order to minimize measurement biases due to different initial footpad thickness, the expressed percentages of PHA-induced swelling responses (%) were calculated as described by Brown et al. (2015) for cane toads (Rhinella marina). 100×PHA thickness at x hour−PHA thickness at 0 hourPHA thickness at 0 hour−Saline thickness at x hour−Saline thickness at 0 hourSaline thickness at 0 hour

Statistics

Data are presented as means ± 1 SEM. Each measurement of PHA swelling response was replicated six times. A very high repeatability of these measurements was observed using intraclass correlation coefficients (r = 0.99, P < 0.001). Therefore, hereafter the mean value of the six measurements of the swelling response for each individual was used for analyses.

Differences among group means were analyzed using analyses of the variance and covariance. Assumptions of ANOVAs and ANCOVAs were tested and when not met, the nonparametric Kruskal–Wallis H test was used. Tukey’s HSD test was carried out to evaluate pairwise comparisons after ANOVAs, and Dunn’s multiple comparisons test was performed after Kruskal–Wallis H. Species, sex and PHA assay effects on body mass were evaluated using a mixed-design ANOVA with an autoregressive (order 1) covariance structure (Littell, Pendergast & Natarajan, 2000) including pre and post-assay body masses, sex and species as fixed factors and subjects nested in sex and species as random factors. A mixed-design ANOVA with an autoregressive (order 1) covariance structure was performed to test differences within individuals of the swelling response at different times (6, 12, 24, 48 and 72 h) after the PHA injection in the three species studied, using time after injection and sex as fixed factors and subjects nested in sex as a random factor. The maximum skin swelling response (PHAmax) after the PHA injection of an individual was defined as the highest of the five values measured along trial time (i.e., 6, 12, 24, 48 and 72 h), and the time when PHAmax was reached, as the time of peak. PHAmax was contrasted among the three species using an ANOVA, using full factorial design with taxa and sex as predictor variables.

Likelihood comparisons of the Akaike Information Criterion (AICc) with a second-order bias correction values produced by models using different parameters (i.e., time-after PHA injection and sex as variables and mean body mass as a covariate) for the different species studied were used as tool to select the predictors that described the best model for PHA swelling response within species (Burnham & Anderson, 2004). Using this procedure, variables included in the models were time-after PHA injection and sex. The same process was used to select the predictor variables, taxa and sex, to analyze PHAmax. Since times of peak and transformed values did not meet the normality assumption (D’Agostino’s  K2 test, P < 0.05), a Kruskal–Wallis H test was used to test differences among taxa and sex. The correlation between PHAmax and timing of the peak was analyzed using the Spearman’s rank correlation coefficient. In all tests, statistical significance was accepted for α < 0.05. Statistical analyses were performed using JMP Pro 12.2.1.

Results

Effects of PHA test, sex and species on body mass

A mixed-design ANOVA was conducted to test PHA assay effect on body mass in male and female individuals of the three species studied. Results showed significant body mass pre- and 72 h post- PHA injection changes within subjects (Table 1). Pairwise comparisons showed that all groups, but Asiatic toad males lost body mass after the PHA assay (Fig. 1). Between subjects’ effects (i.e., sex and taxa) and all interactions were statistically significant (Table 1). Sex differences of body mass before the assay were apparent for Asiatic toads, while Dark-spotted frogs and Mongolian toads did not differ. After the assay, body mass differed significantly between sexes in all species (Fig. 1). Body mass differences between taxa were also apparent, Dark-spotted frogs were lighter than Mongolian toads and Asiatic male toads, and in turn, lighter than the female Asiatic toads (Fig. 1).

Figure 1 Effects of the phytohemagglutinin (PHA) assay, sex and taxa on body mass in the three anuran species studied: Asiatic toads, Dark-spotted frogs and Mongolian toads.

Bars represent mean body mass ± 1 S.E.M. Body masses of female and male individuals of each species before the PHA injections are represented by solid gray and black bars respectively, and 72 h postinjection body masses of females and males are represented by unfilled gray and black bars, respectively. Bars or groups of bars connected by either solid (pre vs. post PHA injection), dashed-double-dot (sex) or dashed (species) lines indicate significant differences of body masses by Tukey’s HSD test (P < 0.05).

PHA-induced skin swelling response

Skin swelling response after PHA injection showed significant differences within individuals of the three species studied (Table 2, Ps < 0.0001). All species increased the thickness of the footpad area following the PHA injection reaching a maximum skin swelling response (Tukey’s HSD test, P < 0.05) and then the swelling significantly declined to a minimum at 72 h (Fig. 2). A closer inspection of PHAmax and its occurrence over time (Fig. 3) showed that both parameters varied within species: the 95% CI of PHAmax and times of peak ranged from 18.58 to (29.75% and 10.6 to 19.72 h, 9.766 to 20.56% and 7.738 to 14.01 h, and 21.97 to 31.78% and 17.39 to 23.94 h for Asiatic toads, Dark-spotted frogs and Mongolian toads, respectively. Sex effect or interaction time x sex were not significant in the three species (Table 2). Between species differences in the magnitude of PHAmax were significant (ANOVA, taxa effect: F2,49 = 4.1664; P < 0.022, Fig. 3A), but not for sex (F1,49 = 0.4994; P = 0.484) or taxa x sex interaction (F1,49 = 0.6600, P = 0.522). Tukey’s HSD test showed that maximum PHA-induced swelling response of Dark-spotted frogs was significantly thinner than that of Mongolian toads, and Asiatic toads PHAmax was not different from Mongolian and Dark-spotted frogs (Fig. 3A). Timing of PHAmax peaks between species was also statistically significant (Kruskal–Wallis test: H2 = 12.2179, P = 0.0022, Fig. 3B) with score means of 23.33, 17.5 and 33.89 for Asiatic toads, Dark-spotted frogs and Mongolian toads, respectively. Dunn’s multiple comparisons test showed that Dark-spotted frogs reached PHAmax significantly faster than Mongolian toads, and Asiatic toads had an in-between value, not different from any of the other two species (Fig. 3B). The relationship between PHAmax and timing of the peak between data of all individuals was significant (Spearman’s rank correlation coefficient, ρ = 0.327, P < 0.05), but we fail to find a significant correlation (P > 0.05) within species with or without sex as a category.

Table 1 Results of a full factorial mixed-design ANOVA with an autoregressive (order 1) covariance structure (Littell, Pendergast & Natarajan, 2000) run to test changes within subjects of the body mass before and after the PHA test using assay, sex and taxa as fixed factors and the subjects nested in sex and taxa as random factors.

Source	d.f.	F	P	
Assay	1, 43.3	146.952	<0.0001	
Sex	1, 44	7.876	0.0074	
Taxa	2, 43.9	83.365	<0.0001	
Sex × assay	1, 43.6	8.767	0.0049	
Sex × taxa	2, 43.9	13.683	<0.0001	
Taxa × time	2, 43.4	13.107	<0.0001	
Sex × taxa × assay	2, 43.5	4.988	0.0112	

Table 2 A full factorial mixed-design ANOVA with an autoregressive (order 1) covariance structure (Littell, Pendergast & Natarajan, 2000) was run to test changes of the swelling response after the PHA injection using time after PHA injection, sex and taxa as fixed factors, and the subjects nested in sex and taxa as random factors.

Sex and taxa were included as predictor variables and body mass excluded as a covariate based on likelihood comparisons of AICc of the different models (Burnham & Anderson, 2004).

Species	time after PHA injection	sex	time × sex	
	d.f.	F	P	d.f.	F	P	d.f.	F	P	
Asiatic toads	4, 53.5	11.42	<0.0001	1, 54.4	1.07	0.306	4, 53.9	0.49	0.746	
Dark-spotted frogs	4, 37.9	8.18	<0.0001	1, 14.7	1.69	0.214	4, 39	1.92	0.127	
Mongolian toads	4, 46	31.80	<0.0001	1, 38.9	0.21	0.651	4, 46.5	1.41	0.246	

Figure 2 Mean (±1 S.E.M.) phytohemagglutinin (PHA)-induced skin swelling response, expressed as the percentage of the initial footpad area thickness corrected by the swelling that produce an injection of the same amount of saline solution in the left footpad (see Materials and methods), at 6, 12, 24, 48 and 72 h after PHA injections in Asiatic toads (A), Dark-spotted frogs (B) and Mongolian toads (C).

Points not connected by the same lower case letter are significantly different (Tukey’s HSD test, P < 0.05).

Figure 3 Mean values ±95% C.I. of (A) maximum phytohemagglutinin-induced skin swelling response (PHAmax) and (B) the times of peak of Asiatic toads, Dark-spotted frogs and Mongolian toads.

Bars connected by brackets are significantly different (P < 0.05) by Tukey’s HSD test and Dunn’s multiple comparisons test in (A) and (B), respectively.

Discussion

Consistent with our predictions, male and female anurans responded similarly to PHA (Table 2), suggesting that PHA assay can be used as an effective index for measuring intra-population variations in immune functions within either sex. In addition, PHA assay as an integrative response is comparable in ecoimmunological studies of amphibians with different body sizes, early or delayed primary PHA max may match with their reproductive activities and parasite infections from outdoor environmental conditions.

Effects of the PHA assay, sex and taxa on body mass were marked in the three anuran species (Table 1). All groups except for Asiatic toad males lost body mass after the PHA assay (Fig. 1), suggesting that it is necessary to measure their body masses at least prior to injection (at 0 h) and at 72 h postinjection. Compared with the toe webbing of large-bodied cane toads (Brown, Shilton & Shine, 2011) and the lower leg below the knee of several small-bodied tree frogs (Clulow, Harris & Mahony, 2015), the footpad site injected also displayed a measurable inflammation response after PHA antigen injection, which can be further applied to other anuran species with body sizes in a range of at least 30 to 90 g. As suggested from bird (Smits, Bortolotti & Tella, 1999) and later from amphibian (Josserand et al., 2015) studies on PHA measurement, a more simplified protocol without saline injection should be considered in future research in amphibians.

In the present study, no sex differences were found for immune responsiveness to PHA in the three anuran species (Table 2, Fig. 2). Similar results had been recorded in chinstrap penguins, Pygoscelis antartica (Moreno et al., 1998), wild rodent species (Zhang, Qiu & Wang, 2011; Merlo, Cutrera & Zenuto, 2014), and several anuran species (Brown et al., 2015; Clulow, Harris & Mahony, 2015). The PHA inflammation assay uses a lectin to elicit localized inflammation that reflects an organism’s capacity to mount a dynamic innate and cell-mediated immune response. In a wild rodent, immune responsiveness to primary PHA was positively correlated with the proportion of neutrophils and serum bactericidal capacity in circulating blood (Zhang & Zhao, 2015), and the energetic cost of mounting this response was low (Zhang, Qiu & Wang, 2011; Merlo et al., 2014). In amphibians, the primary PHA injection may mainly involve innate phagocytes and granulocytes, and a second PHA injection would strongly activate lymphocytes (Brown, Shilton & Shine, 2011; Fites et al., 2014) (Josserand et al., 2015). In the present study, the activation of the innate immunity induced by a single dose of PHA antigen may represent male and female anurans responses to cope with infectious diseases and changing environmental conditions. During April and May, Asiatic toads almost terminated their reproductive activities (Ji et al., 1995; Zou et al., 1996), whereas Mongolian toads (Zhou, 1996) and Dark-spotted frogs just initiated their reproductive activities (Wu, 1965, Cao et al., 2000; Jin, Qu & Zhang, 2014; Jin & Zhang, 2014). Male serum testosterone concentrations are higher during the reproductive than in the non-reproductive period in Dark-spotted frogs (HU, 1990), or before reproduction then after reproduction in Asiatic toads (HU, 1989). Based in life history theory, it can be hypothesized that males may show a reduced immune response and an increased susceptibility to disease than females due to the suppressive effects of androgens on the immune system (Zuk & McKean, 1996). Hence, a suppression or at least a reduction of male immune responses is a reasonable prediction if a trade-off relationship exists between immunity and reproduction (Lochmiller & Deerenberg, 2000). Moreover, male anurans might face a greater risk exposed to more infectious diseases than females because they need croak and move to find a mate during reproduction, which is energetically costly too (Hawley & Altize, 2011). Strikingly in this study, despite the different reproductive status, vocal and locomotor activity, no apparent dissimilarities were found between male and female anurans in the immune responsiveness pattern to PHA (Fig. 2). Sex differences in immune functions may be associated to the immunological parameters (Boughton, Joop & Armitage, 2011; Downs, Adelman & Demes, 2014) or vertebrate life history characters involved (Stoehr & Kokko, 2006). For example, semi-terrestrial salamanders, Desmognathus ochrophaeus females were reported to heal slower than males from an experimental cutaneous wound, used as a measure of an integrative immune response, and the difference in healing time was related to potential energetic constraints (Thomas & Woodley, 2015). However, it has also described that Bufo americanus females and larger males display similar proportions of heterophils acting as first-line defenses against bacterial infection in their leukocyte profiles (Forbes, McRuer & Shutler, 2006).

In a range from fast to slow life span rodents, empirical evidences show that different components of the immune system might display similar or different changing tendencies (Martin, Hasselquist & Wikelski, 2006; Martin, Weil & Nelson, 2007; Previtali et al., 2012). As an integrative response, the temporal patterns of the PHA response might change in different rodents, and individuals that responded faster had a lower maximum response than those with a more delayed response (De Bellocq et al., 2006). In twelve Lark species, environmental proxies of antigen exposure explained variation in immune investment better than proxies of pace of life (Horrocks et al., 2015). In anurans, unlike endotherms and regardless of body size, the primary response to PHA antigen injections is mainly attributed to innate immunity of immune system and recognized as the first-line defense to outdoor pathogens (Brown, Shilton & Shine, 2011). The interspecific differences of the magnitude or timing of PHAmax were observed in the three anuran species (Table 2, Fig. 3), suggesting that a trade-off relationship between intensity and latency of the PHA response may hold among anurans. In adult anurans, PHA max may be reached at 12 h (Clulow, Harris & Mahony, 2015), 14 h (Josserand et al., 2015), 24 h (Brown, Shilton & Shine, 2011), or 48 h (Clulow, Harris & Mahony, 2015) after PHA injection. For post-metamorphic amphibians, a range of ecologically meaningful environmental stressors, such as pond desiccation (Gervasi & Foufopoulos, 2008), different protein concentrations of diets (Venesky et al., 2012), and low and high population density (Clulow, Harris & Mahony, 2015) have great influences on the inflammation response induced by PHA antigen. Asiatic and Mongolian toads, and Dark-spotted frogs are sympatric common species, but the two formers are terrestrial and disperse slowly, and the latter inhabit lentic ponds and disperse fast by swimming or jumping (Fei et al., 2009a; Fei et al., 2009b). It has been reported that the helminth and parasite infection rates in Dark-spotted frogs and Asiatic toads are 90.38% and 100%, respectively, and the average intensity of infection is 20.21 and 28.34, respectively in Shanghai (Zhu et al., 2014; Zhao et al., 2014). However, Mongolian toads were not infected by multiple digenetic trematodes from north China (Li & Gu, 1978). PHA-triggered inflammation was similarly impaired by Eimeria sp. infection alone or co-occurring with a number of gastrointestinal nematodes in the subterranean rodent, Ctenomys talarum (Merlo, Cutrera & Zenuto, 2015). In the present study, Dark-spotted frogs with an early PHA response did show a lower peak response than Mongolian toads with a late response (Fig. 3), suggesting that parasite infections may evoke the changes of PHA response patterns. However, in Xenopus laevis, swelling induced by single injection of PHA was not significantly affected by the fungus disease, Batrachochytrium dendrobatidis supernatants, but swelling caused by a secondary injection of PHA was significantly reduced by B. dendrobatidis supernatants (Fites et al., 2014). It is noted that the magnitude of PHAmax was significantly positively correlated with the timing of PHA max between data of all individuals from the three anuran species, but not for the data within species, suggesting that the magnitude and timing of PHAmax is species-specific.

Overall, we found that immune responsiveness to PHA did not differ between sexes for any anuran species (Table 2, Fig. 2), but PHA max and its timing for toads and frogs varied among species (Fig. 3). This scenario allows PHA assay to be applied to a broad range of amphibian species under field conditions, as a simple, inexpensive and effective proxy to quantify and evaluate a facet of immune response. For future ecoimmunological studies of amphibians, PHA assay will play an important role by integrating the changes of other components of immune system facing newly emerging pathogens or environmental factors. Lastly, we caution that it is still necessary to test the validation of PHA assay in advance for a new species by using appropriate internal controls before carrying out studies under specific circumstances.

Supplemental Information

Supplemental Information 1 Original data for body mass and footpad thickness

Original data for the changes of body mass prior to injection (0 h) and 72 h after injection, as well as the changes of the swelling response at 0 h, and 6, 12, 24, 48 and 72 h after injection.

Click here for additional data file.

Thanks to Xinxin Niu, Zhenfei Xian, and Shuli Huang from Anhui Agricultural University for help in the field.

Additional Information and Declarations

Competing Interests

Author Contributions

Animal Ethics

Data Availability

The authors declare there are no competing interests.

Zhiqiang Zhang conceived and designed the experiments, performed the experiments, analyzed the data, contributed reagents/materials/analysis tools, wrote the paper, prepared figures and/or tables, reviewed drafts of the paper.

Chenchen Jin conceived and designed the experiments, performed the experiments, analyzed the data, wrote the paper, prepared figures and/or tables, reviewed drafts of the paper.

Kangshan Qu performed the experiments, analyzed the data, reviewed drafts of the paper.

Enrique Caviedes-Vidal conceived and designed the experiments, analyzed the data, wrote the paper, prepared figures and/or tables, reviewed drafts of the paper.

The following information was supplied relating to ethical approvals (i.e., approving body and any reference numbers):

Animal Care and Use Committee of Anhui Agricultural University provided full approval for this experimental research (Permit number: 20120410).

The following information was supplied regarding data availability:

The raw data has been supplied as a Supplementary File.

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
