# Peer review of "Immune responsiveness to phytohemagglutinin displays species but not sex differences in three anuran species"

_PeerJ, doi:10.7717/peerj.3181_

## Round 0.1 · original submission · Major Revisions

· Academic Editor

Major Revisions

Given that the initial reviews were a bit conflicting, I tried to find additional reviewers and ended up with the situation that your paper has received 4 reviews. They provide a number of very thoughtful suggestions and I expect your revision will address them accordingly. In particular, please have your text revised for grammar and style, preferably by a native English speaker.

Reviewer 1 ·

Basic reporting

No comments

Experimental design

No comments

Validity of the findings

No comments

Additional comments

The paper represents an important contribution for the field in the comprehension of PHA dynamics within and between species. There are a feww suggestions about some specific points that shoud be considered.

Annotated reviews are not available for download in order to protect the identity of reviewers who chose to remain anonymous.

Reviewer 2 ·

Basic reporting

The English is not clear. There are many places where the incorrect word was chosen, sentences are awkwardly structured, and there are almost no articles used throughout the manuscript (the, an, a, etc.)

The intro and background do show context, however, the reference to global climate change in the abstract and introduction are not warranted. There is no research here related to global climate change at all.

The structure conforms to the PeerJ standard and discipline norms.

The figures are relevant, high quality, well labeled & described.

The raw data is supplied

Experimental design

The research is original, primary, and within the scope of the journal.

The research question is mostly well defined, relevant, and meaningful. It is stated how the research fills an identified knowledge gap in sex-specific differences, but it is misstated that it fills a gap in climate change biology.

It is a rigorous investigation performed to a high technical & ethical standard.

The methods are described in sufficient detail and information to replicate.

Validity of the findings

The results are valid and the rational & benefit to literature is clearly stated except for the reference to climate change.

The data is robust, statistically sound, and controlled, although the sample size is on the small side.

The conclusion is wells stated, linked to the original research question, and limited to supporting results.

All speculation regarding the implications of these findings for climate change responses should be removed or explicitly described as speculative.

Additional comments

I think the title should say: ‘but not sex differences' instead of 'but no sex differences'

In the abstract, the phrase 'increased footpad thickness before' does not make sense given that the increase could only occur after the treatment. Please rephrase this.

A phrase that is used in multiple places is 'differ in both sexes'. This should be changed to 'differ between sexes' throughout the manuscript.

The closing phrase of the abstract does not make sense nor does it fit what was found in this study. First, the results did not indicate that male and female anurans adopt unique responses, it was found that there was no unique response between the sexes, and that they responded the same.

This statement (at the end of the abstract) is not supported by the data or information provided in this manuscript:

"suggesting that the multiple patterns of PHA response may be related to their appearance timings of post-hibernation and habitats (land or pond), early or delayed PHA response may reflect their capacities to cope with changing environmental conditions and increasing pathogen threats in the context of global climate changes."

I recommend that this entire section be removed and that the authors consider a concluding statement for the abstract that is backed by empirical evidence and not entirely on unsupported speculation.


L100 At what taxonomic level do these species differ? Order? Family?

L210 Given that other sources provide evidence for a role of androgens in altering male behavior and immune defenses, it would be useful to note the reproductive status of the anurans in this study at the time of collection - were they breeding, was it pre-breeding, post-breeding? - and how that may or may not affect results.

L229 Here the authors giive the year and person who described that species and the authors do the same for their study species, but there are many other places in the manuscript where a species is mentioned without the accompanying credit to those who described the species or the year. I recommend consistency throughout the manuscript.

The following are a few of the edits that are necessary to clear up the English in this paper. This is not all-inclusive, and I recommend that the authors have someone proofread this manuscript to make the English more clear.

L47 ‘are variable, providing plasticity in the response to…”
L49 ‘energetically costly’
L55 used
L60 barabensis
L61 circulating
L65 still some debate
L69 when
L77 was examined
This statement does not make sense:
which are still extremely scarce in amphibians by using of consensus methodology
(Clulow, Harris & Mahony, 2015), especially for those anurans living in different life
environments.


L104 ponds fields
L105 latest for
L106 Please use the term 'sex' instead of 'gender'
L122 tap water
L145 the repeat was very
L188 between sexes
L250 ponds fields space after fields
The phrase 'Timings of post hibernations' is not correct, please rephrase this.
L263 between sexes

Reviewer 3 ·

Basic reporting

Basic Reporting
For the most part the manuscript is clearly written. However there remain
numerous minor grammatical errors or passages where the meaning is unclear. I've included many examples from the introduction, but several other minor errors occur throughout the remainder of the manuscript.

L38. It is not clear what 'it' refers to in this sentence. Maximum PHA response? A difference between sexes?

L44. " their appearance timings of post-hibernation ". Change to "the timing of their emergence from hibernation"

L54. change "cost" to "costly"

L56. Change "preproduction" to "reproduction"
L60. Change " widely techniques" to " widely used techniques"
L65. Change "Cricetulus Barabensis" to " Cricetulus barabensis"
L67. Change " observed an" to "observed as an".
L71. Change " are still some debating" to "is still some debate".
L72. Change " responsiveness" to "response".
L73. Change "of species specificity" to "species specific".
L98. Remove "life".
L102. Remove "classification".
L109 Replace ; with .
L127. Change "tape" to tap".

Experimental design

Some additional information on Methods and Results are necessary to help the reader understand the study and the interpretation of findings.

L.120-4 How many sites were each species collected from? Just one site for each? Were the Asian and Mongolian toads collected from the same site and the dark-spotted frogs from a different site?

L156. Was this two different analyses? One on 0hr mass and another on 72hr mass? Or was it a single analysis done on the change in mass between 0hr and 72hr?

L174. Shouldn't there be two tables here? One for 0hr and one for 72hr mass? What is the PHA Test effect? Is this the 0hr vs 72hr effect? This analysis is unclear. Does each individual appear twice in the analysis? Once at 0hr and once at 72hr? It must, since Error df = 94 even though there are only a total of 53 animals. If this is the case the analysis is pseudo-replicated. Animal ID needs to be included as a random effect if each individual has two observations.

L195. All the time-related effects seem to be missing from the repeated measures table. For instance, there needs to be a significant time*species interaction affect (that indicates that the way the PHA response varies over time differs among the three species) before 2-way analyses for each species is done separately. If the time*species interaction is nonsignificant there is no need to pursue the separate 2-way analyses.

Validity of the findings

L115 Why just hibernation timing? There are many life history traits that could affect immune response. What about age at maturity litter size, longevity or growth rate?

I don't think it is useful to try to link differences in PHA response to differences in hibernation time. The three species no doubt differ in many, many ways that could impact their immune responses. It seems banal to arbitrarily choose this single difference as a likely causative factor.

Also, if each species was collected from a single site, rather than from several sites, technically the immune differences could be site-specific, rather than species.

Similarly, there are no data presented on measures of different environmental stressors among the species. In fact, one could argue that environmental stressors are equally important to all species.

No measures were made of parasite load, pathogen pressure, ambient temperature, nor of any relevant life history trait. It would be better to focus on the fact that species-specific differences in immune response exist. Trying to tie those differences to unmeasured environmental or ecological differences isn't very enlightening.

Additional comments

This is a useful and carefully conducted study and is generally well presented. I made several comments above, regarding additional information to include. The species-specific immune responses could have arisen through many mechanisms from phylogenetic constraint to site or population-specific factors. Trying to isolate hibernation timing or environmental stressors as likely causes, without supporting evidence, seems unwarranted.

·

Basic reporting

The manuscript proposed by Zhang et al. for publication in PeerJ reports the results of their investigation of temporal variation of phytohaemagglutinin (PHA)-induced cutaneous inflammation in three anuran species from China. PHA skin-swelling test is one of the most common methods in ecological immunology utilised to measure inflammatory responsiveness. This article is one of few studies on this topic in amphibians and given its comparative nature the findings are of a potential interest. The article is clear, logically structured and well written. I have, however, several concerns.

Experimental design

The article would be much more powerful if it expanded the methodology above the basic description of the swelling response. You could have tested whether PHA is an appropriate stimulant in amphibians on a first place – it has been show in mammals, for instance, that PHA is not the optimal stimulant for this type of a research since there is lack of association between swelling and cellular infiltration into the tissue (Bílková et al. 2016). Is there any evidence available showing in amphibians that the metrically measured swelling response induced by PHA is indicative of (correlated to) the processes (cellular accumulation, cytokine expression) that are going on inside of the tissue? This should be clearly mentioned in the text since otherwise it is not clear what aspect of the immune response is being measured.

It would be also good if the haematological state of the individuals (that is indicative of health) was investigated. A priori blood cellular composition has been shown to affect the tissue swelling after PHA stimulation (Bílková et al. 2015). It might have been difficult in the small species but in the medium-sized and large-sized frogs this could have been done I believe.

Validity of the findings

One of my major concerns in this manuscript is that some of the interpretations appear to be too simplified. At ln 63 you write: “more swelling may represent a ‘better’ or ‘stronger’ integrative immune response”. I agree that larger tissue thickness change indicates stronger response but I disagree that this is indicative of a better response. It is not clear better to what – evidence accumulated showing that greater immune response does not necessarily mean better protection against infection diseases or higher survival probability (Graham et al. 2005). Greater inflammation means more damage to the tissues. Greater response to PHA is also in some cases indicative of an ongoing infection (Vinkler et al. 2012). As pointed out by Demas et al. (2011), “examining isolated immune components such as PHA, although still informative, do not provide a comprehensive measure of the immune system and should not be interpreted as such.”. Immune responses to different antigens differ remarkably. PHA is not a pathogen-derived protein and its ability to mimic naturally occurring pathogens has been questioned repeatedly (Owen and Clayton 2007). I can see much benefit out of the application of this method but it must be always clear what is to be measured, i.e. inflammation, and why.

Additional comments

Minor comments
- Ln 53: “… select appropriate immune parameters…” – appropriate to what? This is dependent on which question you aim to answer.
- Ln 57: similarly as in my previous comment – you need to state validity for what application you are talking about.
- Ln 130: Why did you decide to apply PBS injection as a control? A simplified protocol to measure the immune response to PHA (instead of injecting PBS into the other leg the magnitude before and after the injection of PHA is measured) was proposed by Smits et al. (1999) in birds and later by (Josserand et al. 2015) for amphibians.
- Ln 148: instead of a three-way ANOVA with PHA test as a factor I would suggest to calculate a linear model with PHA swelling response as a continuous variable.
- Ln 173: I am not sure about the phrasing of this sentence and the sentences with a similar structure below.
- Figure 2: I may have missed it somewhere but how did you test the differences between the categories – I did not find any mention of some post-hoc test. Was this calculated using the Tukey test?
- Ln 188-190 and Figure 3: I am not sure here what this is meant to tell. I do not see any information additional to the one presented earlier (Fig. 2). If this is important, please better explain the meaning of the analysis or (if redundant) exclude it from the manuscript.
- Ln 267: “specific immune response” – I do not understand what kind of specificity is being discussed – definitely not “antigen-specific” I believe.
- Supplement Table S1: I think each individual should have its own code otherwise the reader cannot be sure that each line represents an independent individual.

Literature

Bílková B, Albrecht T, Chudíčková M, Holáň V, Piálek J, Vinkler M (2016) Application of Concanavalin A during immune responsiveness skin-swelling tests facilitates measurement interpretation in mammalian ecology. Ecol Evol 6:4551-4564
Bílková B, Vinklerová J, Vinkler M (2015) The relationship between health and cell-mediated immunity measured in ecology: Phytohaemagglutinin skin-swelling test mirrors blood cellular composition. Journal of Experimental Zoology Part A: Ecological Genetics and Physiology 323:767–777
Demas GE, Zysling DA, Beechler BR, Muehlenbein MP, French SS (2011) Beyond phytohaemagglutinin: assessing vertebrate immune function across ecological contexts. Journal of Animal Ecology 80:710-730
Graham AL, Allen JE, Read AF (2005) Evolutionary causes and consequences of immunopathology. Annual Review of Ecology Evolution and Systematics 36:373-397
Josserand R, Troianowski M, Grolet O, Desprat JL, Lengagne T, Mondy N (2015) A phytohaemagglutinin challenge test to assess immune responsiveness of European tree frog Hyla arborea. Amphibia-Reptilia 36:111-118
Owen JP, Clayton DH (2007) Where are the parasites in the PHA response? Trends in Ecology & Evolution 22:228-229
Smits JE, Bortolotti GR, Tella JL (1999) Simplifying the phytohaemagglutinin skin-testing technique in studies of avian immunocompetence. Functional Ecology 13:567-572
Vinkler M, Schnitzer J, Munclinger P, Albrecht T (2012) Phytohaemagglutinin skin-swelling test in scarlet rosefinch males: low-quality birds respond more strongly. Animal Behaviour 83:17-23

---

## Round 0.2 · Minor Revisions

· Academic Editor

Minor Revisions

Although one of the reviewers is happy with the revised version, the other reviewer identified a number of small issues that have to be addressed. However, I'm confident that you'd be able to incorporate these suggestions relatively easily.

Reviewer 3 ·

Basic reporting

fine

Experimental design

fine

Validity of the findings

fine

Additional comments

I am satisfied that the authors have addressed the comments I made on an earlier version of this manuscript. I think it is more clearly written, and the interpretations more conservative. I think it can be published in its present form.

·

Basic reporting

The revised version of the manuscript is much improved. I would, however, suggest shortening the abstract that is now very long (e.g. given that your data are size-normalised I do not think it is necessary to write “As for taxa, Dark-spotted frogs were lighter than Mongolian toads and Asiatic toad males, and in turn, lighter than the females of Asiatic toads.” in the abstract – “differing in size” may be enough). In some cases I also do not think that you reflected the point of my earlier comments (please see the Comments for the author section).

Experimental design

No comments

Validity of the findings

No comments

Additional comments

Abstract and also lns 242-244 - I do not understand the numbers. You write: “Relative maximum swelling response (PHAmax) and its occurrence over time ranged from 18.58 to 29.75 % and 10.6 to 19.72 min (12 h postinjection)”. But the PHAmax could not occur in 10-20 minutes after stimulation. I believe the 12 hours after infection are correct. Or do I understand it wrongly? Perhaps “min” should have been “mm” referring still to size of the swelling.
Ln 70 – please omit “reflecting tissue damage” – inflammation involves (causes) tissue damage but it would be erroneous to describe inflammation as a process that “reflects” tissue damage. By the way, inflammation is definitely one of the cases of an “integrative” response – it involves many types of immune cells. But in immunology you would not use “integrative immune response” since this is very unspecific description that may mean almost everything.
Ln 71 and Ln 80 – the meaning of the references (Graham et al. 2005) and (Vinkler et al. 2012) supporting my previous remark that greater immune response / inflammation does not necessarily mean better protection against infection diseases or higher survival probability was not reflected in the new version of the manuscript. You do need to admit that the more may not be the better but it may well be the opposite! This is crucial since this simplified and erroneous view is still very common in ecology.
Ln 74 – It would be fair to mention that despite reasonable effort the histological base of the swelling response to PHA was not supported in several mammalian studies, e.g. Merlo et al. (2014).
Ln 76 – the association of PHA skin swelling to leukocyte profiles in blood has been also reported in birds – see Bílková et al. (2015).
Ln 101 – actually, we observed the peak of the swelling response to PHA in mice as soon as 3 hours after the injection (Bílková et al. 2016).
Lns 111-112 – this is the trouble – there is no general “simple and reliable method, like PHA test, to evaluate the immune system responsiveness”. That is why we argue so intensively against simplifications in interpretation of immunological data. The PHA test refers to some rather general pro-inflammatory capacity but does not tell you anything about the capacities of other immunological mechanisms. Please see the argumentation in Vinkler and Albrecht (2011) – you do not need to cite the paper if you do not want to; my point is, please read it and formulate your statements accordingly. I understand that you yourself may be aware of the weaknesses in the generalisation of your statements but your readers may not. There are many researchers in zoology who do not have virtually any immunological background and only use the test since it has been used and cited before. Please do contribute to rectification of the traditional interpretational errors in ecoimmunology.
Ln 128 – although technically speaking correct, point i) is rather trivial – do you really wish to mention this?
Lns 244-250 – It seems to me that the information provided on lns 244-247 is somehow repeated (with statistics) on lns 247-250. In results this appears redundantly ad the remark on test power should be included into discussion.
Ln 254 – Given the measurement intervals - would not medians (instead of means) provide more appropriate comparisons?
Ln 287 (legend to Fig. 2) – I do not understand “corrected by the amount of liquid injected (see Materials and methods)” – from Materials and Methods ln 150 it appears clear that all individuals received the same dose. Or am I wrong?
Lns 326-328 – I am afraid I also do not understand the sentence “male and female anurans may mainly rely on the activation of the innate immunity induced by a first dose of PHA antigen, which is used to cope with infectious diseases and changing environmental conditions”. In your experiment there was only a single PHA injection – or not?
Ln 388 – I am not sure if you can interpret this as the necessity for large sample sizes – I would be inclined to believe that this results from interspecific variation (described in this study – trend across species) not number of individuals within species.

Replies to referee
- I do not understand the sentence “Our results provide an available injection site for amphibians with similar body sizes by applying PBS injection as a control.” How is the site of the injection related to the fact that you apply PHA injection only or both PHA and PBS? I do believe that the protocol suggested by Smits et al. (1999) for birds and later by (Josserand et al. 2015) for amphibians with only one injection is more considerate to the welfare of the animals and should be applied.


Literature

Bílková B, Albrecht T, Chudíčková M, Holáň V, Piálek J, Vinkler M (2016) Application of Concanavalin A during immune responsiveness skin-swelling tests facilitates measurement interpretation in mammalian ecology. Ecol Evol 6:4551-4564
Bílková B, Vinklerová J, Vinkler M (2015) The relationship between health and cell-mediated immunity measured in ecology: Phytohaemagglutinin skin-swelling test mirrors blood cellular composition. Journal of Experimental Zoology Part A: Ecological Genetics and Physiology 323:767–777
Graham AL, Allen JE, Read AF (2005) Evolutionary causes and consequences of immunopathology. Annual Review of Ecology Evolution and Systematics 36:373-397
Josserand R, Troianowski M, Grolet O, Desprat JL, Lengagne T, Mondy N (2015) A phytohaemagglutinin challenge test to assess immune responsiveness of European tree frog Hyla arborea. Amphibia-Reptilia 36:111-118
Merlo JL, Cutrera AP, Zenuto RR (2014) Inflammation in response to phytohemagglutinin injection in the Talas tuco-tuco (Ctenomys talarum): implications for the estimation of immunocompetence in natural populations of wild rodents. Canadian Journal of Zoology 92:689-697
Smits JE, Bortolotti GR, Tella JL (1999) Simplifying the phytohaemagglutinin skin-testing technique in studies of avian immunocompetence. Functional Ecology 13:567-572
Vinkler M, Albrecht T (2011) Handling immunocompetence' in ecological studies: do we operate with confused terms? Journal of Avian Biology 42:490-493
Vinkler M, Schnitzer J, Munclinger P, Albrecht T (2012) Phytohaemagglutinin skin-swelling test in scarlet rosefinch males: low-quality birds respond more strongly. Animal Behaviour 83:17-23

---

## Round 0.3 · accepted · Accept

· Academic Editor

Accept

I am happy with the final modifications.